# Synchronization in collectively moving inanimate and living active matter

Michael Riedl [1] ✉, Isabelle Mayer[1], Jack Merrin[1], Michael Sixt[1] ✉ & Björn Hof[1] ✉

Whether one considers swarming insects, flocking birds, or bacterial colonies, collective motion arises from the coordination of individuals and entails the adjustment of their respective velocities. In particular, in close confinements, such as those encountered by dense cell populations during development or regeneration, collective migration can only arise coordinately. Yet, how individuals unify their velocities is often not understood. Focusing on a finite number of cells in circular confinements, we identify waves of polymerizing actin that function as a pacemaker governing the speed of individual cells. We show that the onset of collective motion coincides with the synchronization of the wave nucleation frequencies across the population. Employing a simpler and more readily accessible mechanical model system of active spheres, we identify the synchronization of the individuals' internal oscillators as one of the essential requirements to reach the corresponding collective state. The mechanical 'toy' experiment illustrates that the global synchronous state is achieved by nearest neighbor coupling. We suggest by analogy that local coupling and the synchronization of actin waves are essential for the emergent, self-organized motion of cell collectives.

Collective motion emerges from the self-organization of individuals and occurs in living as well as inanimate active matter across a wide range of length scales. Observations reach from macroscopic flocks of birds, or schools of fish, over the coordinate motion of swarms of robots[1–3], down to the collective motion of cells[4] and synthetic self-propelled particles[5–9]. While the type of interactions and adjustments that give rise to the emergent order vary from case to case, individuals always must converge to closely matched speeds. The most prominent theoretical model capable of reproducing e.g., flocking reduces the phenomenon's complexity to a balance between the alignment strength between individuals and the noise disturbing the system[10]. As in this and similar minimal models, the speed of individuals is assumed to be constant and uniform, the effects of velocity fluctuations, which can destabilize the ordered state[11,12], received little attention. The assumption of uniform velocities is not valid for most biological systems and fluctuations in speed across individuals can potentially prevent the ordered state of a flock from arising[12]. Especially in densely packed environments, like monolayers of migrating cells, feedback mechanisms between individuals are essential to ensure the

coordination of the individuals' velocities, which is a key requirement for the motion of the collective.

Here, we show that uniform velocity in collectively moving populations arises through the synchronization of an oscillating propulsion mechanism. This principle is independent of the exact nature of the underlying oscillators and their mode of coupling. We demonstrate this concept based on two systems, the biochemical oscillator underlying actin waves in endothelial cells and for the mechanical oscillating motors in active spheres. Extending on these findings, we further test the sensitivity of the collectively moving state to perturbations caused by uncooperative individuals and by disrupting the mechanisms that gave rise to it in the first place.

## Results

### Synchronizing actin waves frequency onsets collectivity
We tracked the motion of cultured endothelial cells adhering to two-dimensional adhesive patterns shaped as disks or rings. When cells were plated at low density and had no contact with their neighbors, they moved without any preferential direction[13]. When grown to

---

[1]Institute of Science and Technology Austria (IST Austria), Klosterneuburg, Austria. ✉e-mail: michael.riedl@ist.ac.at; sixt@ist.ac.at; bjoern.hof@ist.ac.at

confluency, cells transitioned from an initially chaotic motion to a coherently moving state, where the collective adapted a carousel-like motion with uniform locomotion speed of the individual cells (Fig. 1a)[13–15]. Polymerization dynamics of the actin cytoskeleton provide the driving force for cell locomotion. We thus visualized actin dynamics by genetically introducing a fluorescent actin reporter in endothelial cells[16]. Time-lapse fluorescent imaging revealed waves of high actin density traveling through cells (Supplementary Movie 1), a phenomenon previously described in other cell types and a consequence of the excitable dynamics of the actin polymerization machinery[17–22]. The formation of waves was evident in single cells as well as in collectives (Supplementary Movie 1–3). We initially confined single cells within a cell-sized circular adhesive

pattern to quantify wave dynamics (Supplementary Movie 1). From the resulting kymographs, we extracted individual waves propagating with an average speed of $9.4 \pm 2.7\,\mu m\,min^{-1}$ (Supplementary Fig. 1a–f). To assess the interplay between the internal actin waves and the locomotion speed of the single cell, we restricted cells to one-dimensional movement on line-like adhesive patterns (Supplementary Movie 2). The nucleation frequency of newly arising waves within moving single cells varied from 2.2 to 5.5 $h^{-1}$ and correlated positively with the speed of the migrating single cell (Fig. 1d, e, Supplementary Fig. 1g, h). That correlation was consistent with observations in other cell types where the nucleation frequency of actin polymerization waves correlated with the velocity of the leading edge of a cell[17,22–25].

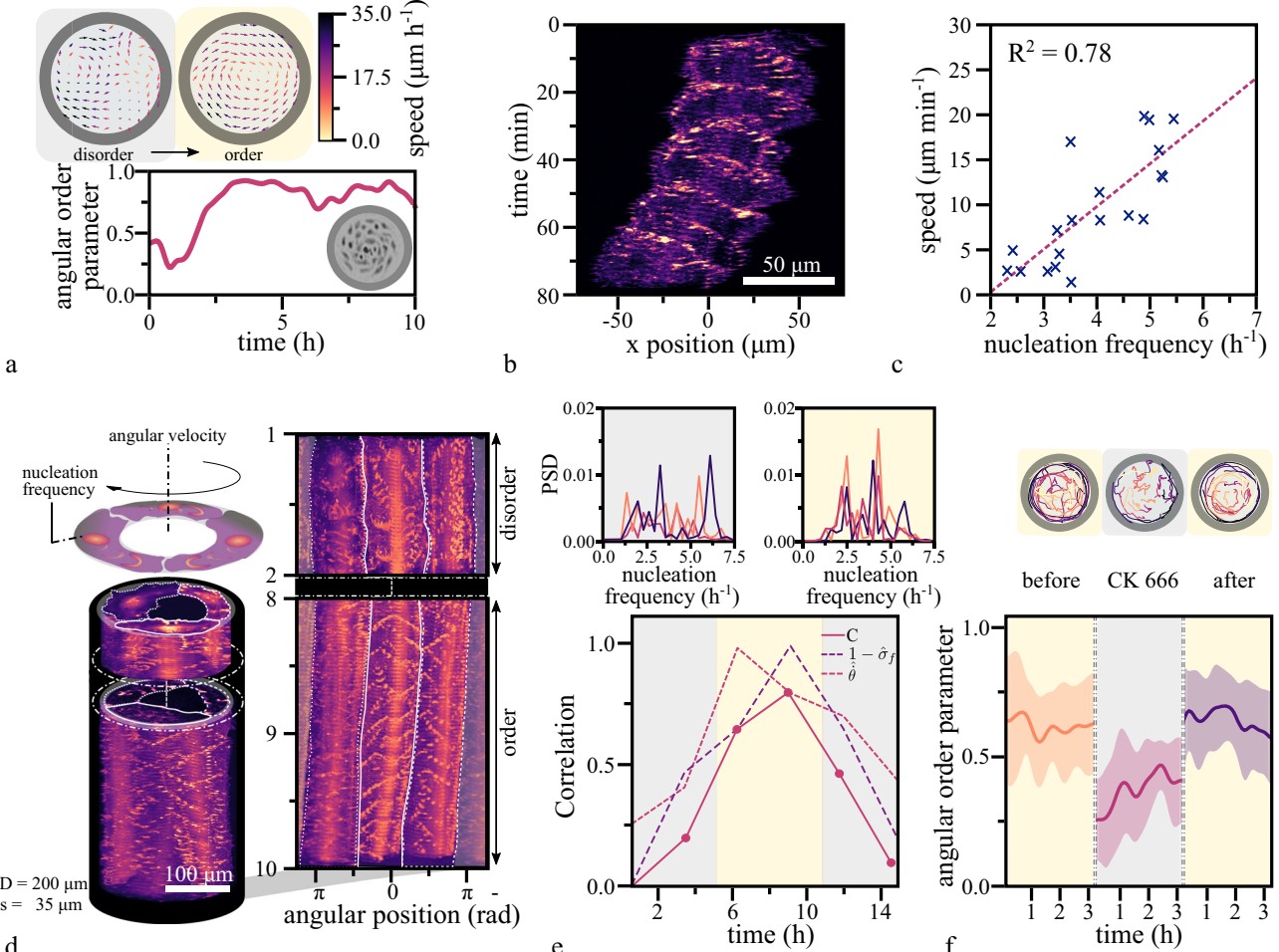

**Fig. 1 | Transition to order in confined endothelial cells. a** Top: The state of disorder (gray shaded region) and order (beige shaded region) is illustrated by the corresponding velocity field of the confluent endothelial cell layer on a circular pattern (D = 200 μm). Bottom: The steep increase in the angular order parameter ($\Phi = \frac{1}{N}\sum_{j=1}^{N} \hat{v}_j\,\mathbf{e}_{\varphi j}$) quantifies the transition from the initially disordered to the ordered motion, where $\hat{v}_j$ denotes the normalized velocity and $\mathbf{e}_{\varphi j}$ the unit vector in the azimuthal direction of agent cell j in a population of size N. **b** The space-time representation of a single cell moving on an adhesive line pattern (width = 50 μm). **c** By plotting the mean migration speed of a single cell against the nucleation frequency of actin waves, we find a positive correlation (n = 20). **d** Top: A schematic representation of the collective migration of endothelial cells on a ring-shaped adhesive pattern. Below: The space-time representation of a representative subset of the acquired time series of a configuration of 3 cells on a ring (diameter = 300 μm, width = 35 μm), before and after the onset of collective rotation. Adjacent: In the corresponding polar transform, the frequency locked state becomes evident by the uniform vertical spacing between the periodically reappearing polymerizing

actin waves with respect to the individual cells and across the collective. The white lines indicate the boundary between individual cells. **e** Bottom: The correlation parameter C characterizes periods of small standard deviations in frequency and high mean velocities across the population. During phases of ordered, collective migration (beige-shaded region), the correlation parameter C approaches values close to unity and decreases towards zero while the system is in disorder (gray-shaded regions). Top: The corresponding power spectrum density depicts the dominant frequencies present. The main nucleation frequency of polymerizing actin waves is around 4.8 $h^{-1}$, and the first harmonics are present at around 2.4 $h^{-1}$. **f** Top: Representative cell trajectories are shown during individual stages of a CK666 treatment experiment. During the presence of the ARP2/3 inhibiting drug, cells are actively migrating, but the global order is lost. After removing the drug, the carousel-like motion recovers. The transitions between states can be quantitatively identified by the drop and consecutive rise in the angular order parameter (n = 10) (Mean ± SD).

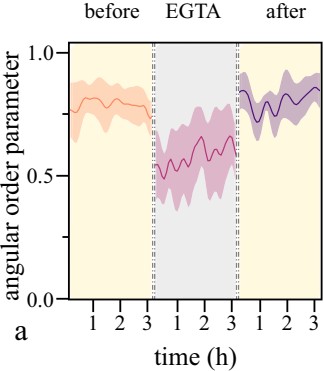
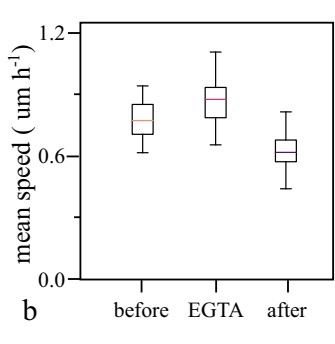
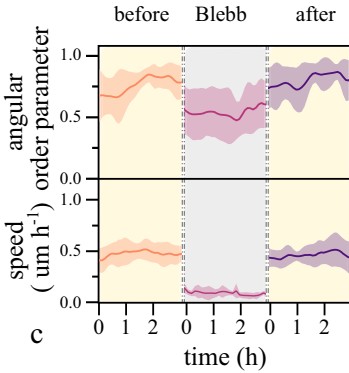

**Fig. 2 | Destabilizing the order in both systems. a** Depleting the medium of calcium disrupts cadherin-based junctions and renders collectively rotating cell patterns uncoordinated. The subsequent replenishment of the medium recovers the collective rotation. ($D = 150 \, \mu m$, $n = 10$) (Mean ± SD). **b** In the disrupted state single cell migration speed is increased. ($n = 10$) The bottom and top of each box represent the first and third quartiles, respectively, while the band inside the box marks the median. Whiskers extend to 1.5 times the interquartile range. **c** Top: Inhibition of myosin activity renders patterns less coordinated. Bottom: Concurrently, single-cell migration speed reduces, comes to a halt, and recovers after replenishing the medium. ($n = 8$) (Mean ± SD).

To study the dynamics of actin waves during the transition to collective migration, we altered our experimental setup from a two-dimensional circle to a one-dimensional ring. In this configuration, cells are restricted to interact with exactly two adjacent cells, enabling us to observe how these nearest neighbor interactions influence actin dynamics. On ring patterns containing three cells, individual nucleation frequencies were uncorrelated during periods where cells were not rotating collectively (Fig. 1d, e, Supplementary Movie 3). With the onset of collective migration and for as long as it persisted, the nucleation frequencies adjusted to a common frequency around 4.8 h⁻¹ (Fig. 1e). In non-rotating patterns, the nucleation frequency of polymerizing actin waves remained non-uniform across the collective (Supplementary Fig. 2). By introducing the correlation parameter $C = (1 - \hat{\sigma}_f) \cdot \dot{\hat{\theta}}$, we quantified the correlation between the global maximum in the normalized mean angular velocity $\dot{\hat{\theta}}$ and the global minimum in the normalized standard deviation of the frequencies $\hat{\sigma}_f$ of the collective. We normalized both quantities by their respective global maxima. We found that C reached its maximum during phases of ordered collective migration and decreased when collective motion ceased. Thus, the onset of collective motion coincided with the synchronization of nucleation frequencies (Fig. 1e).

## Perturbations of collective motion

To probe whether collective locomotion and actin waves are causally related, we inhibited the chemical reaction controlling the waves of polymerizing actin. Traveling actin waves are known to be driven by the self-activating WAVE protein complex[20]. WAVE drives nucleation of actin polymerization via the Arp2/3 complex, and polymerized actin inactivates the WAVE complex, thus creating an excitable system[18–20]. We pharmacologically perturbed Arp2/3 mediated actin nucleation with the drug CK666. We found that at a 100 μM concentration, the Arp2/3 inhibitor eliminated actin waves while single-cell locomotion was maintained, albeit at lower velocities[17]. The carousel-like rotations on adhesive surfaces were abrogated upon treatment with CK666, and cells migrated in an uncoordinated manner instead. Upon washout of the drug, the effect was reversed, and collective rotation was restored (Fig. 1f, Supplementary Movie 3)[14].

Although the transition of confined cells to the rotating state occurs reliably, it is highly sensitive to perturbations targeted toward disrupting cellular locomotion or nearest neighbor interactions. For instance, the loss of cadherin-based junctions compromises the populations' ability to establish the ordered rotation[14,26]. Extending this notion, we investigated whether cell-cell junctions are required not for establishing but for maintaining collectivity. After the establishment of the rotating state, we disrupted cadherin-based junctions by depleting the medium of calcium using EGTA. Contrary to the sustained collective rotation in ring-like confinements[14], we observed that in circular confinements, the rotating collective turned into uncoordinated migrating individuals subsequent to treatment (Fig. 2a, Supplementary Movie 3). We attribute this behavioral difference between confinement geometries to the one-dimensional nature of ring patterns. In a ring, the degrees of freedom in migration are limited and therefore collective migration requires a lower level of coordination. The subsequent replenishment of the medium reverted both effects (Fig. 2b, Supplementary Movie 3).

Next to its role in trailing edge retraction, myosin II-mediated actomyosin contractility is involved during the formation and strengthening of cadherin dependent junctions[27]. To see whether inhibition of myosin II increases sensitivity to perturbations, we treated collectively rotating cells with 80 μM Blebbistatin. Initially, the treatment led to a gradual loss of coordination and, eventually, the arrest of cell motility (Fig. 2c, Supplementary Movie 3). After replenishment of the medium, the collective rotations reemerged.

Overall, our data suggest that collectively moving cells are coupled chemical oscillators driven by the excitable dynamics of actin polymerization within each cell[17–20]. We propose that intercellular coupling through nearest neighbor interactions gives rise to frequency locking, which in turn allows cells to adopt a uniform speed, enabling collective motion. Conceptually, the process of synchronization is independent of the precise nature of the underlying coupling mechanism. However, in the case of polymerizing actin waves, their emergence is a result of chemical oscillations. Any coupling mechanism that influences their behavior is likely to involve a chemical component. Given the responsiveness of actin polymerization to mechanical stimuli[28], we hypothesize that the coupling could be of a mechano-chemical origin. Although the sensitivity of the emergent collective motion to perturbations of the potential coupling mechanisms is evident, the precise nature of the underlying coupling remains to be understood.

## Synchronization onsets collectivity in motorized spheres

The observed collective migration that emerges from the synchronization of the individuals' oscillators is reminiscent of recent theoretical models reproducing swarm formation[29,30]. One such model introduces agents, known as swarmalators[29], whose velocity is expressed as a function of the phase of an internally oscillating process. By further introducing coupling between the oscillators, they potentially synchronize and the collective transitions from an uncoordinated motion to a swarm-like state. To experimentally test if nearest neighbor coupling suffices to establish synchronization as an underlying mechanism to unify velocities and eventually lead to collective motion, we

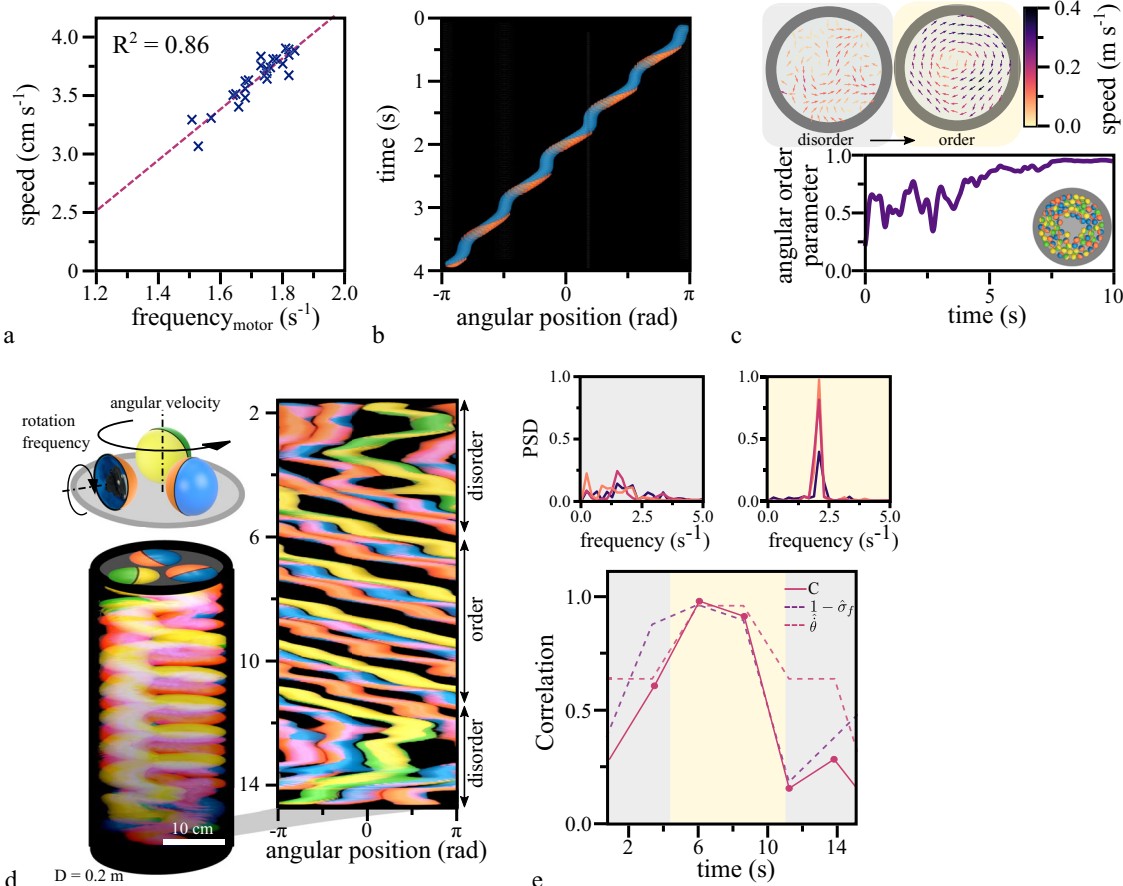

**Fig. 3 | Transition to order in confined motorized balls. a** The mean speed of a single motorized ball correlates positively with the frequency of the internal motor, which was measured in an opened ball while the shell was fixed (*n* = 25). **b** The staircase-like trajectory in the space-time representation visualizes the oscillatory motion of a single ball in a circular confinement. **c** Top: A representative velocity field of the motion of a population of motorized balls (*N* = 73) in a circular confinement (D = 0.9 m) in the state of disorder (gray shaded region) and order (beige shaded region). Below: The angular order parameter quantitatively shows the spontaneous transition from the initial disordered to the ordered motion quantified. **d** Top: The schematic depiction of a system containing three motorized balls.

Below: The space-time representation of a subset of the acquired images. Adjacent: In the corresponding polar transformation, the synchronized state is recognizable by the periodically repeating, parallel paths of the ball trajectories. The slope indicates the uniform speed within the collective. **e** Top: Plotting the power spectrum density in both states illustrates the transition between disorder and order. The initially broad distribution of frequencies converges to defined peaks around 1.9 s⁻¹. Below: The correlation parameter C reaches values close to unity in the synchronized state, where the mean velocity is a maximum, and the standard deviation of the frequency maxima reaches a minimum.

introduced a model system of mechanical 'toys' with comprehensive degrees of freedom. We employed 'weaselballs' (D.Y. Toy), autonomously moving spherical shells driven by an internal, unbalanced motor freely rotating around an axis connecting the poles of the shell. The motor applies a constant torque and while the shell is fixed, causing the continuous revolution of the internal mass (Supplementary Fig. 3a). We extracted the frequency of the revolutions by tracking the position of the freely moving motor over time and found that it varies between individual balls (Fig. 3a, Supplementary Fig. 3b, c). When the motorized balls were released on a sufficiently rough substrate, they rolled with oscillating speed along unstable trajectories (Fig. 3b). After coming into contact with the boundary of the circular confinement, a single ball traced it, a behavior common for active particles (Supplementary Fig. 2b)[1,31]. Along the resulting circular trajectory of the motorized ball, the speed oscillated (Supplementary Fig. 3d). The frequency of the oscillating speed was reduced with respect to the frequency of the corresponding freely rotating motor (Supplementary Fig. 3e, f), indicating that additional forces e.g., friction or gravity acting on the shell and the motor influenced the frequency of revolution.

When we placed multiple balls in circular confinements, their dynamics transitioned from a disordered motion to a smooth and ordered collective rotation pattern (Fig. 3c, Supplementary Movie 4). Once established, the rotational movement for large numbers of balls remained stable. In contrast, a confinement containing three randomly selected balls alternated between ordered and disordered states, allowing us to analyze the transition between these states (Fig. 3d, Supplementary Movie 4). When we computed the angular velocity of each ball with respect to the center of the circular confinement, we found that the velocities oscillated out of sync during periods of disorder and in sync during the ordered state. Additionally, during the period of the ordered state, the mean angular velocity of the population $\dot{\theta}$ showed a maximum. This maximum overlapped with a minimum in the standard deviation of the instantaneous frequencies $\hat{\sigma}_f$ of the oscillators. We quantified this correlation with the previously introduced correlation parameter *C* and found that it reached values close to one during periods of synchronized motion and strongly decreased when the system was in disorder (Fig. 2e). This shows that the onset of collective motion coincides with the locking of velocity oscillations.

**Synchronization emerges at the level of the driving mechanism**
Synchronization requires coupling between the individual oscillators[32,33]. In our system, coupling arose from the contact between balls. To investigate the extent of the coupling, we extracted the

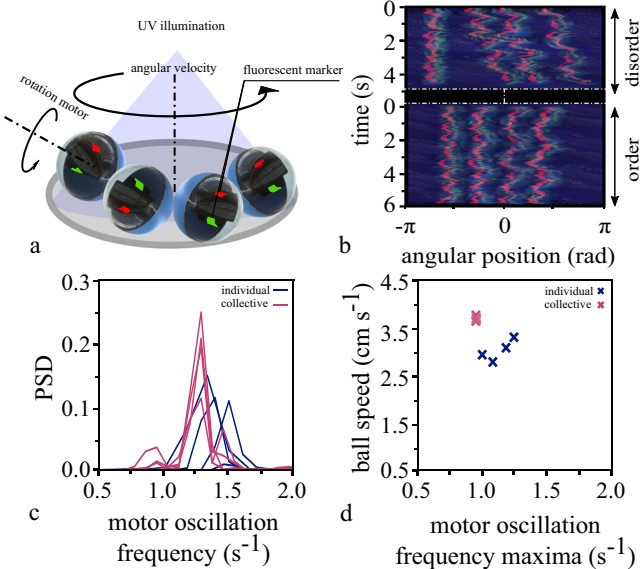

**Fig. 4 | Experimental setup and synchronization of the motors in confined balls with transparent half-spheres. a** The schematic representation of the experimental setup using 4 balls with transparent half-spheres in a circular confinement (D = 0.4 m). We attached fluorescent tape to the internal motors and illuminated the setup with UV light to visualize the motion of the internal motor. **b** We show the space-time representation of a disordered and ordered period in the comoving reference frame with respect to the leading ball. The red oscillations of the fluorescent tape correspond to the oscillation of the internal motors. During periods of order, the oscillations synchronize their frequencies and phases. **c** The corresponding power spectrum density shows this change in the frequency distribution of the internal motors' oscillation. **d** The oscillations of the internal motors become independent of the balls' motion by stabilizing it with respect to the velocity corresponding to the individual ball's center of mass. Plotting the frequency maxima of these oscillations shows that the frequencies lock during synchronized movement.

frequencies of the oscillating speed from trajectories of single balls individually moving in confinement. We found a distribution of frequencies, between 1.3 and $1.9 \, s^{-1}$ (Supplementary Fig. 3e). The same population of balls, when put together in a circular or annular confinement, transitioned to a coherently rotating state. In this state, the initially wide frequency distribution collapsed to a single, common frequency (Supplementary Fig. 4). This finding raised the possibility that the coupling mechanism did synchronize not only the phases of oscillating motion of rolling balls but also the frequencies of the motors. To measure this, we produced a set of balls with optically transparent half-shells and directly tracked the motion of the motor independent of the outer shell (Fig. 4a, b). From the oscillations of each motor, we extracted the frequency spectra (Fig. 4c), revealing that the motor frequencies indeed locked by converging to a common frequency (Fig. 4d). Thus, the observed synchronization of the collective extends to the level of the force generating mechanism of the individual balls.

### Collective motion is sensitive to uncooperative individuals

We next investigated mechanisms to destabilize the ordered state or to prevent it from arising. We probed this by continuously increasing the number of balls collectively moving within a confinement (D = 0.9 m). After reaching a critical number of agents (N = 82) in the confinement, the ordered motion of the now densely packed population became unstable, transitioned to the disordered state and the ordered state did not re-emerge. The transition to disorder corresponded to a sudden drop in the angular order parameter (Fig. 5a, Supplementary Movie 5). This would suggest that the destabilization is caused by the increased density of balls in the system. However, when the population

of balls was preselected to have matching natural frequencies to avoid variability within the collective, the ordered state still emerged in close to fully packed confinements (Supplementary Movie 5). This indicated that the stability of the ordered state depended on the distribution of frequencies rather than the density of the population.

As shown previously[34,35], collective motion is sensitive to uncooperative individuals or variations within the population. In flocks, already a low fraction of non-aligning agents can disrupt the collective movement[34–36]. To see if this holds true for our systems, we first probed our system of motorized balls by exchanging active with inactivated balls by switching their motors off. The substitution of about 4% of the population with uncooperative balls led to the breakdown of the collective behavior, in quantitative agreement with values found in numerical simulations on flocking[34] (Fig. 5b, Supplementary Movie 5).

To see whether our findings on destabilization in our system of motorized balls are translatable to our system of migrating cells, we aimed for conceptually analogous experiments. In agreement with our observations in the system of motorized balls, previous findings showed that the rotating motion of migrating cells in confinement is arrested after reaching a critical density threshold[15]. To perform the experiment analogous to introducing uncooperative individuals, we chose to treat endothelial cells with the actin polymerization inhibitor MycB at doses that inhibit actin wave formation but not adhesion and spreading on the substrate. Contrary to most other pharmacological agents, MycB binds covalently to actin and, therefore does not wash out or leak into other cells upon dilution[37]. The treatment of a cell population with 0.5 μM MycB rendered individual cells barely migratory and in circular confinements unable to establish collectivity. (Supplementary Fig. 5a). We pre-incubated a 10% subpopulation of endothelial cells with MycB and simultaneously labeled it with a fluorescent dye (TAMRA) so that it could be identified and tracked when mixed with untreated cells. To exclude any unspecified effects caused by the fluorescent dye used, we observed a homogenous population of stained cells under identical experimental conditions. These cells displayed no difficulties establishing the collective rotation (Supplementary Fig. 5b). MycB treatment renders the subpopulation of treated cells into passive individuals that were dragged around by their active neighbors. The presence of these uncooperative individuals prevented the ordered state from arising, which was in agreement with our observations in the mechanical experiment as well as models simulating flocking[34] (Fig. 5c, Supplementary Movie 5).

### Discussion

In nature, propulsion mechanisms are often based on repetitive processes like the flapping of wings or fins or the rotation of cilia. For the collective motion to arise and persist, feedback mechanisms are required to enable these autonomous oscillators to adjust[38,39] and converge to a uniform speed[36]. The original theoretical work on swarmalators proposed that weak coupling between the underlying oscillators is a sufficient and conceptually simple way to establish the necessary feedback between individuals[29]. In both, the biological and the mechanical systems we introduce here, the underlying oscillator and the resulting propulsion mechanism are not identical in their nature. Yet, the emerging synchrony yields uniformity in speed and thereby improves the effectiveness of the coordinated migration.

In swarming collectives, the process of weak coupling is usually not established via a global or swarm-wide communication, but through local nearest neighbor interactions which propagate information across the collective[40]. Nearest neighbor interactions provide a reliable communication method in the absence of global information and were shown to suffice as a design for self-organizing artificial swarms[2]. The speed as well as the robustness of nearest neighbor information transfer limits the size of the coherently moving population[35,41]. This limitation potentially explains the similarity in length scales of collectively rotating cells observed here and those

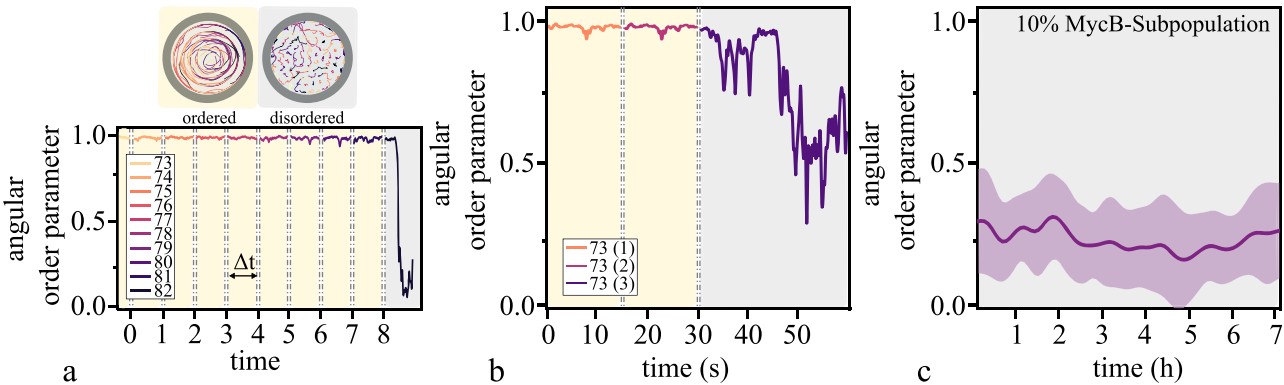

**Fig. 5 | Destabilizing the order in both systems. a** We approached the critical number of agents at $n = 82$ by incrementally increasing the number of balls in a circular confinement (D = 0.9 m). Top: Representative trajectories are displayed for both states over an equal time interval of 2 s. They show the transition from a collectively moving state to a Brownian-like motion. Bottom: The steady states' angular order parameter fluctuates close to unity for a representative time interval of $\Delta t = 25$ s. The transition to disorder is evident by the sudden decrease in the angular order parameter when reaching a critical number of agents within the confinement. **b** Consecutively rendering 4% of the population of balls inactive leads to the breakdown of the order (3 inactive balls within a collective of 73 balls). **c** Partially inactivating actin polymerization in a subpopulation of cells prevents order from arising ($n = 12$) (Mean ± SD).

found in spontaneously emerging transient vortices appearing in larger confinements or unconfined monolayers of otherwise chaotically migrating cells[13]. As we show, the propagation process can be compromised by variations between the oscillators or the presence of uncooperative individuals, which destabilize the arising order. This sensitivity poses a constraint on the emergence of collective motion, restricting the variability between individuals and the number of uncooperative entities present.

While the overall concept is not restricted to the cellular or mechanical agents studied here[42], it remains to be seen whether the collective motion of more complex organisms relies on similar principles.

## Methods

### Lentivirus production
The lentivirus production was performed, as previously described[20]. In brief, LX-293 HEK cells (Clontech) were co-transfected with Lenti-CRISPRv1 or LentiCRISPRv2, packaging- psPAX2 (Addgene no. 12260) and pCMV-VSV-G envelope plasmids using Lipofectamine 2000 (Thermo Fisher Scientific) as recommended by the manufacturer, and cells were resuspended in R10 medium 1 day before transfection. The supernatant was collected after 72 h and stored at −80 °C.

### Cell culture
Human Aortic Endothelial Cells (HAoEC), obtained from PromoCell GmbH, were cultured in a growth medium (Endothelial Cell Growth Medium MV, PromoCell). Cells were grown and maintained in a humidified incubator at 37 °C and 5% $CO_2$ and passaged after reaching 70% confluency. During long-term imaging, we added 1% penicillin–streptomycin antibiotic (Invitrogen) to the growth medium.

### Cell infection and selection
For the infection of HAoEC with the Lifeact-GFP reporter construct[16], we use a 70% confluent monolayer in a T-75 flask of a passage number not higher than 2–3. After aspiration of the initial growth medium, we added 10 ml growth medium supplemented with 0.5 ml Lentivirus and incubated overnight. Subsequently, we removed the growth medium and washed the cell layer 3× with PBS before adding fresh growth medium. Afterward, we proceeded by starting the selection process by supplementing 3 μl Blasticidin (Gibco, Thermo Fisher Scientific) from a stock solution (10 mg/ml) to 10 ml cell culture medium. The selection is completed after 4–6 days or until a sufficiently low number of non-fluorescent cells remain.

### Photomasks
The patterns were designed with Coreldraw X8 (Corel Corporation) exported to DXF, then converted to GERBER format with LinkCad. Chromium quartz photomasks (100 mm or 125 mm; PhotoData/JD Photo-Tools) were used.

### Patterns
The adhesive patterns were generated using PLL-g-PEG (Surface Solutions, Switzerland), as previously described[21] with a few changes. Initially, we rinsed the coverslips with ethanol and dried them with filtered compressed air. After activation by exposure to UV light for 15 min, we incubated the coverslips with 0.3 mg/ml PLL-g-PEG for 1 h at 37 °C, which created an inert, non-adhesive coating. Subsequently, UV irradiation through the chromium quartz photomask removes the non-adhesive coating locally, creating the desired pattern. Before seeding the cells, we washed the coverslips 1× with Ethanol and subsequently 2× with PBS and coated them with 0.2% gelatin.

### Microscopy
Throughout the image acquisition process, the cells were within an incubation chamber under a controlled $CO_2$ atmosphere. Timelapse fluorescent images were recorded using an inverted wide-field Nikon Eclipse Ti-2 microscope equipped with a DS-Qi2 monochrome camera and a Lumencor Spectra X light source (475 nm Lumencor). NIS Elements software (Nikon Instruments) was used for acquisition control for e.g., multi-position imaging.

### CK666 treatment
After an initial unperturbed period, we treated the cells by adding a 100 μM solution of the ARP 2/3 inhibitor CK666 in preconditioned growth medium to the sample. Subsequently, the effect was reverted by removing the medium and washing the cells 2× with preconditioned growth medium. Each consecutive phase of this treatment was followed by a stabilization period of 4 h before image acquisition.

### EGTA treatment
Analog to the CK666 treatment, the cells were treated with EGTA. Initially, the cells were left undisturbed for roughly 8 h until they developed the collective rotation. Then, we applied a 2 mM solution of EGTA in preconditioned growth medium to the sample. Prior to the recovery period of the experiment, we removed the treatment solution and gently washed the cells twice with preconditioned growth medium, which led to the reversal of the effect. Each phase of this

treatment was followed by a stabilization period of 4 h before image acquisition.

## MycB treatment and TAMRA staining

We prepared the staining solution by diluting 1 μl of a 10 mM stock solution of TAMRA in 1 ml PBS. Subsequently, we washed the adherent HAoEC monolayer 1× with PBS and added the previously prepared staining solution. After an incubation period at room temperature of 10 min, we washed the cells 2× using preconditioned growth medium. In the next step, we diluted the MycB stock solution to 0.1 μM in the growth medium and added it to an equal amount of growth medium in the flask with the MycB treated cells resulting in a final concentration of 0.05 μM. After a 30 min incubation period, the cells were gently washed 3× with preconditioned growth medium. After a recovery period of 30 min, cells were detached, mixed together with non-treated cells, and seeded on micropatterns.

## Weaselball preparation

We removed the 'weaselballs' (D.Y. Toy) from their shipment boxes and separated them from the attached weasels. The balls consist of two half-spheres which can be screwed open to detach them, and after, one AA battery can be inserted. Before each experiment, the AA batteries had to be collectively exchanged to ensure similarity in the torque generation throughout the populations. Empty or nearly empty batteries can lead to artifacts in the rolling behavior of a ball.

## Production of transparent half-spheres

We used silicone rubber (Wagnersil 22 NF) to create an inverse mold of the outside and inside of the desired half-sphere. Subsequently, we removed the half-sphere from the mold, mixed the transparent epoxy resin adhesive with the hardener (Let's Resin), and filled the resulting gap. Previous to pouring, the two components had to be well mixed and free of air bubbles.

## Image analysis

FIJI imaging processing software (https://fiji.sc/) was used for image and video microscopy processing and analysis.

## Trajectory tracking

For extracting the trajectories of the balls or the cells from recorded movies, we used the 'manual' or 'semi-automated tracking' feature provided by the plugin TrackMate within the imaging process software FIJI[43] We post-processed the extracted trajectories using Python 3.7.

## Phase and frequency extraction for motorized spheres

From the extracted trajectories of the balls, we calculated the angular velocities with respect to the center of the confinement. From these oscillating velocities, we could extract the instantaneous phases and frequencies using the Hilbert transform method provided within the scipy library.

## Frequencies extraction for polymerizing actin waves in cells

For extracting the nucleation frequency from the kymographs generated with FIJI, we corrected the instantaneous position of each cell by the mean angular speed of the population. In the case of ring patterns, this step was precluded by a polar transformation of the initial images. Transforming the images in a cell-comoving reference frame results in a kymograph equal to a stationary pattern.

Subsequently, we use the Fast Fourier Transformation method within the Scipy library to calculate the frequency spectrum at multiple positions within a cell over the time frame of interest. Each position within a cell results in a frequency spectrum, from which the global maximum of the dominant peak value across all calculated frequency spectrums serves as a selection criterion.

## Data availability

The authors declare that the data supporting the findings of this study are available within the paper and its supplementary information files. Imaging data sets generated during the current study are available from the corresponding author on request.

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

## Acknowledgements

We thank K. O'Keeffe, E. Hannezo, P. Devreotes, C. Dessalles, and E. Martens for discussion and/or critical reading of the manuscript; the Bioimaging Facility of ISTA for excellent support, as well as the Life Science Facility and the Miba Machine Shop of ISTA. This work was supported by the European Research Council (ERC StG 281556 and CoG 724373) to M.S.

## Author contributions

M.R., M.S., and B.H. conceived the experiments and wrote the manuscript, with critical feedback from all authors. M.R. designed, performed, and analyzed the experiments, with the help of I.M., M.R., and J.M. designed the mask used for patterning. M.R. wrote the image analysis scripts.

## Competing interests

The authors declare no competing interests.
