## [Peer Review File · Nature Communications]

Synchronization in collectively moving inanimate and living active matterREVIEWER COMMENTS

Reviewer #1 (Remarks to the Author):

The article by Riedl and colleagues studies the coupling of actin contractile wave in collectively migrating endothelial cells. In endothelial cells, actin waves propagate at the surface while cells are migrating. When a group of cells migrate in a non coordinated manner, each cell has its own frequency and phase of wave nucleation. By forcing the cells to adopt a collective migration behaviour, essentially by confining the cells onto an adhesive ring that forces the cells to rotate, the authors show that not only they coordinate their movement, but they also correlate their actin waves. They further show that the coupling induces both a synchronization of the phase and the frequencies of waves in different cells. By disrupting the actin waves, the authors show that the coupling of actin waves is required for the collective migration behaviour. They propose that the coupling of chemical oscillators is essential for collective cell migration, and that this coupling occurs by mechanical contact between cells, rather than signalling.

To test their hypothesis, they undertake an original approach, which should be acknowledged. They confined oscillating spherical motors in rings and observed when their motion correlated, and when the activity the motor (inside the ball) are correlated. Interestingly, they find that the density of motors mildly affects the propensity to collectively move, but that above a certain density, or below another one, the collective migration is abrogated, supported the mechanism of physical coupling of collective motion. Overall the paper is nicely written, and the strategy original. The hypothesis is novel, and clearly demonstrated for the ball motors. However, while I am in principle in favour of publication, I wish that the analogy between the cellular system and the motor system shall be tested further.

1-The origin of the coupling of the waves remains unclear. Of course, the analogy with the motor system suggests that it is purely mechanical, but the authors have not completely ruled out that adhesion molecules such as cadherin, also involved in cell-cell interaction. Could the authors perform experiments with cells where cadherins or adhesive molecules have been knock-down or knock out? On the reverse, it could be interesting to see how increased cell adhesion through overexpression of cadherins could act on the coupling.

2-One of the limitations of the pharmacological approach used by the authors to disrupt actin waves also probably affect motility, since the drugs used affect actin turnover. Would it make sense to use other ways? For example, inhibiting myosins using blebbistatin could be useful. Also, 100 μM CK666 sounds a lot to only partially perturb Arp2/3. Is that a common concentration? Also, maybe perturbing Calcium signalling could be another way to actin waves without affecting too much the acto-myosin machinery.

3-On the other side, the balls have very little interactions between them, except direct contact. I was wondering if the authors could perform the same kind of experiments in a water bath, or silicon gel bath, to see how coupling provided by the viscous drag between balls affects the coordination of the collective motion.

Reviewer #2 (Remarks to the Author):

This work will be of interest to those studying cell biology, developmental biology, and cancer biology as well as those interested in emergence and self-organization in active matter.

In this paper, Reidl et al. address the role of velocity fluctuations in generating collective order. The work uncovered an effect of mechanical feedback on coordinating velocities across units in the system, leading to collective motion. Using an established tool to label actin, the authors illustrate actin waves and spirals that have been previously noted but these movies represent an advance and are quite striking. They then find that collective cell motion coincides with the onset and duration of synchronized frequencies of actin nucleation. Pharmacologically perturbing actin nucleation caused cells to migrate in an uncoordinated manner. A cleverly selected mechanical model (weaselballs) substantiates the mechanism that syncing of velocity oscillations drives collective motion. Using both biological and inanimate systems, the authors make novel connections and present a compelling mechanism: that cell-cell coupling gives rise to frequency locking, allowing cells to adopt a uniform speed, which leads to collective motion. The unique integration of a novel physical model with high-quality live-cell imaging adds to the work's broad appeal.

I considered whether this paper presents too much interesting data for one manuscript. Given the range of models and concepts, I could envision at least 2 papers from the work here. This is represented in the fact that each Supp Movie consists of multiple "Acts". That said, these "Acts" are well put together and labeled. Further, the results are related and the links are well explained. The data are clearly presented and the conclusions of this paper are well supported by data. I am in support of publication and only have relatively minor comments:

Comments:

Figure 2/3: When the authors placed "multiple balls" in circular confinements, they observe the system transitions to ordered collective rotation. With three balls, the system toggles between order and disorder. Above 82 balls, order can be lost. What is the minimum number of balls where the system displays consistently ordered rotations? Are there similar constraints for cell number?

Figure 4c, Movie 5: This last "Act" of Movie 5 is likely meant to be compared to the last "Act" of Movie 3 but it would be best to have a control for Movie 5 with no treated cells but cells that have also been dyed.. The authors should also describe in more detail how they determined that the treated cells were "passive individuals that were dragged around by their active neighbors." Tracking/quantification would be appropriate here.

Figure 1 and Figure 4c: It is not clear why collective cell migration assays are performed sometimes in rings and sometimes discs. The authors should explain the rationale for each of these geometries. Also the authors should consider repeating the results in Fig. 4c in rings where the 1D nature would allow for a more controlled forum to test the effect of a disruptor cell on collective motion.

Reviewer #3 (Remarks to the Author):

In their work, Riedl et al. investigate a novel mechanism that could be responsible for the coordination of collective migration of cells. To this end, they utilize fluorescence microscopy to observe a system of human endothelial cells in circular or ring-shaped confinements. Focusing on the actin cortex which has been labeled with LifeAct-GFP, the authors observe

dynamic waves of actin polymerization. These waves fall in line with observations in previous studies, as they behave like excitation waves, can create spiral patterns and show velocities of a similar magnitude.

The novelty in the present study lies in the fact that the work finds a clear correlation between the synchronization of wave nucleation events and the collective turning motion of cells within a circular or ring-shaped confinement. This suggests an intriguing mechanism which enables cells to coordinate with their nearest neighbors solely due to oscillations in their actin cytoskeleton at shared interfaces. A mechanical – rather than biochemical – coupling process like this has not been described in detail before, and as such provides a valuable and very important contribution to the field.

The experimental data presented by the authors is of high quality and allows for a quantitative analysis of actin wave dynamics and cell turning at the same time, which is not trivial considering the different length- and time-scales of the two processes.

Furthermore, the authors introduce a simplified model consisting of hollow balls that are driven by an internal motor that is rotating an unbalanced weight. This system is used as an analogy, showing the results of an established physical synchronization process. Both studied systems share the same characteristic behaviours, hinting again at a synchronization of actin polymerization waves as a source of collective migration.

Because of these points, I want to recommend this work for publication, as I think that it will be a good fit for this journal. However, before it is submitted to a wider audience, I would like a few clarifications as well.

Major questions and remarks:

Most of them are related to the direct analogy between the motor-driven ball and the cell, which is not an immediately obvious one to make. The ball is directly driven by the unbalanced weight on the motor and rigid. Once there is some sort of confinement that limits its degrees of freedom, there is an immediate feedback from the available range of motion to the possible orientation and spinning frequency of the motor.

In a cell, this is not as straightforward. The waves are generated and deform the outline, but it deforms wildly even in the absence of waves. On top of that, cells can propagate via other means, as the CK666 experiments show. Other studies in different cell types go in a similar direction and identify polymerization waves mostly as a polarizing mechanism for the cell, which would also explain the results in this paper. Once the waves have polarized the cells in the cluster, those would follow the laws of collective motion of classic active particles (assuming e.g. constant and equal speeds), which have also been found to engage in collective turning motion in circular confinements only in the presence of next-neighbor interactions.

1.) In this context, it would be interesting to see not only the standard deviation (STD) of the wave nucleation frequency over time, but also the magnitude of the average wave nucleation frequency in the cluster itself. Does it correlate with the turning rate?

2.) In a similar sense, for a direct feedback, one would expect both the turning rate and 1 minus the normalized STD of the oscillation frequency to peak at the same time, as in Fig

2e. How significant is it that in Fig1e, the frequencies start aligning perfectly about 3 hours after the collective motion has started?

3.) Is it possible to affect other parts of the system with drugs? Could e.g. inhibition of myosin with blebbistatin lead to waves in the absence of collective motion?

4.) Why is specifically the nucleation frequency the quantity that is synchronized, rather than some phase or relative position? Is it connected to the observation in the movie that in the turning state, waves mostly travel “back to front”, while they seem to be more chaotic before the onset of motion? How representative is that notion?

All of this does not affect the validity of the individual results for the synchronization, it is more related to the physical origin of the effect. Currently, there is no clear distinction between coupled oscillators and polarized active particles, or even something more abstract. Depending on this, the model system would be more of a qualitative comparison to illustrate the important characteristics, rather than a simplified equivalent. While doing all of the necessary tests might be out of scope for this particular work, it is still something that should be noted.

Minor remarks:

1.) I would rephrase the introduction of the order parameter C in the main script. “The maximum / the minimum” sounds like it refers to one specific instance during the experiment where the value is at its highest/lowest. A general “maxima/minima” is clearer.

2.) The arrows in figures 1a and 2c could be a bit bigger.

Response to the reviewers

REVIEWER COMMENTS

Reviewer #1 (Remarks to the Author):

The article by Riedl and colleagues studies the coupling of actin contractile wave in collectively migrating endothelial cells. In endothelial cells, actin waves propagate at the surface while cells are migrating. When a group of cells migrate in a non coordinated manner, each cells has its own frequency and phase of wave nucleation. By forcing the cells to adopt a collective migration behaviour, essentially by confining the cells onto an adhesive ring that forces the cells to rotate, the authors show that not only they coordinate their movement, but they also correlate their actin waves. They further show that the coupling induces both a synchronization of the phase and the frequencies of waves in different cells. By disrupting the actin waves, the authors show that the coupling of actin waves is required for the collective migration behaviour. They propose that the coupling of chemical oscillators is essential for collective cell migration, and that this coupling occurs by mechanical contact between cells, rather than signalling.

To test their hypothesis, they undertake an original approach, which should be acknowledged. They confined oscillating spherical motors in rings and observed when their motion correlated, and when the activity the motor (inside the ball) are correlated. Interestingly, they find that the density of motors mildly affects the propensity to collectively move, but that above a certain density, or below another one, the collective migration is abrogated, supported the mechanism of physical coupling of collective motion.

Overall the paper is nicely written, and the strategy original. The hypothesis is novel, and clearly demonstrated for the ball motors. However, while I am in principle in favour of publication, I wish that the analogy between the cellular system and the motor system shall be tested further.

We thank the reviewer for their appreciation of our approach, the detailed comments on our manuscript and interesting suggestions.

1-The origin of the coupling of the waves remains unclear. Of course, the analogy with the motor system suggests that it is purely mechanical, but the authors have not completely ruled out that adhesion molecules such as cadherin, also involved in cell-cell interaction. Could the authors perform experiments with cells where cadherins or adhesive molecules have been knock-down or knowck out? On the reverse, it could be interesting to see how increased cell adhesion through overexpression of cadherins could act on the coupling.

We agree with the referee, the exact nature of the coupling mechanism in the case of the cells is not yet fully understood, despite the substantial effort dedicated to answering this question in recent publications. (Doxzen et al., 2013; Luo et al., 2023). Please note that we

are not claiming that the coupling between cells is purely mechanical. The analogy to the ball system encompasses that the coupling relies on next nearest neighbour interactions which in turn leads to synchronization. To avoid confusion we now emphasize in the manuscript that our analogy between the transition in our motorized ball system and the transition in our cell collectives does not rely on an identical coupling mechanism.

Regarding the role of cell-cell adhesions these indeed plays a crucial part in the coupling between neighbours. The suggested experiments including the knock down of α -catenin which removes cadherin-based adhesions, or expressing the transcription factor Snail1 which impaired cell-cell junctions have previously been carried out (Doxzen et al., 2013; Luo et al., 2023). Similarly in ring patterns, the perturbation of Arp2/3, Rac1, and cadherin-based adhesion have been related to a failure of establishing the rotational motion. (Jain et al. 2020) These findings show that cells subjected to either method are unable to establish coordinated rotation.

However, interpreting these results is not a straightforward matter, as pointed out in the referees' next comment. While both downregulation and knockout of cadherin-based adhesions results in the inability to establish coordinated motion within a cell layer, they also inhibit the motility of each individual cell. Cells compensate the reduced expression of cadherin-based adhesions by strengthening the cell-substrate adhesions. This in turn reduces the motility of individual cells. (Balasubramaniam et al., 2021)

Following the referee's suggestion, we have performed further tests to probe the role of the adhesions in coupling. In this case we investigated the effect of inhibition of cell-cell junctions using EGTA subsequently to the establishment of the collective rotation. Contrary to the previously observed sustained collective rotation after treatment in ring-like confinements (Jain et al. 2020), we find that in circular confinements, subsequent to treatment, the collective rotating population turned into more or less randomly migrating individuals. We attribute this behavioural difference between confinement geometries to the one-dimensional nature of ring patterns that limits the degrees of freedom in migration and therefore requires less coordination. **(Fig. 1a, Response Movie)**

Fig. 1 | Disrupting nearest neighbour interactions

a, Depleting the medium of calcium disrupts cadherin-based junctions and renders previously collectively rotating cell patterns uncoordinated. The subsequent replenishment of the medium recovers the collective rotation. ($D = 150 \text{ um}$, $n = 10$) **b**, In the disrupted state single cell migration speed is increase.

While coordination appeared to be reduced, some but not all confined cell patterns showed an increase in migration speed. An effect that may result from the loss of cell-cell junctions. **(Fig. 1b)**

These findings suggest that the initiation of the collective movement, rather than their maintenance, depends on the establishment of proper cell-cell contacts. (Jain et al. 2020) This result could be explained by the tension induced during the assembly of cell-cell junctions, which we will further address in response to the subsequent comment from the referee.

In conclusion we believe that the coupling is very likely a mechano-chemical process, given its complex nature as evidenced by past studies and our further tests, it unfortunately remains beyond the scope of the present study to fully explain it, but as we pointed out above, the central point of our analogy is the nearest neighbour coupling and not limited to a purely mechanical nature.

2-One of the limitations of the pharmacological approach used by the authors to disrupt actin waves also probably affect motility, since the drugs used affect actin turnover. Would it make sense to use other ways? For example, inhibiting myosins using blebbistatin could be useful.

We in principle agree with the referee and do believe that there is an intricate relationship between the coupling mechanism and mechanisms that influence locomotion in our study. It is important to note that adhesive cells, such as the endothelial cells studied here, rely on myosin for migration. Inhibition of myosin with Blebbistatin is a double-edged sword in addressing the source of coupling. While it can perturb the coupling between cells and therefore perturb collective motion, it also renders adhesive cells immobile and impacts the individual cells' migration.

Any readout that relies on speed of a collective has to be interpreted with care. For example in ring-like confinements, inhibition of myosin eventually leads to a complete halt of collective motion, which cannot be decoupled from the loss of migration in individual cells (Jain, et al. 2020). At the same concentration of Blebbistatin (80 μM) in circular confinements, we find that cell locomotion arrests before an effect in coordination is visible. **(Fig. 2, Response Movie)**

At lower concentrations (40 μM) cell migration eventually also arrest. However, in the transient period it appears that configurations that require less coordination, e.g. a symmetric arrangement, can sustain the collective rotation for longer time periods or until perturbed by an uncoordinated individual. This is in agreement with the interpretation drawn from experiments conducted in ring-like confinements, where the rotations are more robust and the order parameter drops gradually before cell motility ceases. (Jain et al. 2020) The susceptibility of Blebbistatin treated patterns to perturbations resulting from uncoordinated cells during migration aligns well with the involvement of myosin in the formation and strengthening of cadherin dependent junctions. (Hoelzle and Svitkina, 2011)

Also, 100 μM CK666 sounds a lot to only partially perturb Arp2/3. Is that a common concentration?

While the concentration of CK666 at 100 μM appears to be high, it is commonly used in literature, see for example Jain, S., et al. 2020. Subsequently to more than 8 hours of treatment with this concentration, cell collectives recover their migration and appear healthy after washing them with medium.

Also, maybe perturbing Calcium signalling could be another way to actin waves without affecting too much the acto-myosin machinery.

The depletion of calcium renders cells unable to establish collective motion due to its involvement in the formation of junctions in addition to its influence in cell-substrate adhesions and migration. See comment 1, for a more detailed answer.

3-On the other side, the balls have very little interactions between them, except direct contact. I was wondering if the authors could perform the same kind of experiments in a water bath, or silicon gel bath, to see how coupling provided by the viscous drag between balls affects the coordination of the collective motion.

We thank the referee for this very interesting suggestion. Initially, we performed experiments to evaluate the behaviour of a single ball in water. The results strongly depend on the density, viscosity and the height of the liquid. Buoyancy forces change the effective forces acting on the ball and alter its individual rolling behaviour. At sufficient water levels, balls float and are not effectively moving forward.

At water levels that allow balls to roll while in contact with the substrate, their oscillatory motion triggers travelling surfaces waves. These surface waves outpace the ball and

transverse the perimeter of the circular confinement and eventually interact with the ball that initially triggered it. Whether the collision of the wave amplifies or perturbs the balls motion depends on the balls' instantaneous phase at the moment of impact. This interaction is apparent as a dent in the speed of the ball. **(Fig. 3a, Response Movie)**

This complex hydrodynamic interaction and the interaction timing between ball and the waves' impact depends on a multitude of variables. Next to e.g., the diameter of the confinement and the fluid properties, such as density and viscosity, the height of the water level strongly influences the height h and speed of the wave itself. **(Fig. 3a)** The increased viscosity of a glycerine-water mixture leads to a higher dissipation of the injected energy, which suppresses this emergent phenomenon.

The coupling between two individual balls strongly relies on the collisions and friction between them. While collisions cause phase shifts that eventually synchronize their motion, friction leads to converging frequencies. This hydrodynamic interaction adds another layer of complexity to an already intricate interaction.

After addition of another second ball, we adjusted the now raised water level due to the displaced fluid. Still, the pair of balls manages to synchronize via the unavoidable collisions, but over reduced time periods. **(Fig. 3b, Response Movie)** The emitted waves by one ball perturb the motions of the other over a distance. Hence, the balls tend to desynchronize again.

However, hydrodynamic interaction can also give rise to a collective migration pattern different from that in a dry environment. A pair of synchronized balls can establish a vortex flow. This rotational flow can dominate the movement of the balls and drag them along at a constant speed. **(Fig. 3b)** Here, the oscillatory motion of the balls is superimposed with a reduced amplitude onto the constant speed of the fluid flow.

While we agree with the referee that this is an interesting experimental configuration, it clearly adds different physical mechanisms and fundamentally changes the dynamics of the system.

Reviewer #2 (Remarks to the Author):

This work will be of interest to those studying cell biology, developmental biology, and cancer biology as well as those interested in emergence and self-organization in active matter.

In this paper, Reidl et al. address the role of velocity fluctuations in generating collective order. The work uncovered an effect of mechanical feedback on coordinating velocities across units in the system, leading to collective motion. Using an established tool to label actin, the authors illustrate actin waves and spirals that have been previously noted but these movies represent an advance and are quite striking. They then find that collective cell motion coincides with the onset and duration of synchronized frequencies of actin nucleation. Pharmacologically perturbing actin nucleation caused cells to migrate in an uncoordinated manner. A cleverly selected mechanical model (weaselballs) substantiates the mechanism that syncing of velocity oscillations drives collective motion. Using both biological and inanimate systems, the authors make novel connections and present a compelling mechanism: that cell-cell coupling gives rise to frequency locking, allowing cells to adopt a uniform speed, which leads to collective motion. The unique integration of a novel physical model with high-quality live-cell imaging adds to the work's broad appeal.

I considered whether this paper presents too much interesting data for one manuscript. Given the range of models and concepts, I could envision at least 2 papers from the work here. This is represented in the fact that each Supp Movie consists of multiple "Acts". That said, these "Acts" are well put together and labeled. Further, the results are related and the links are well explained. The data are clearly presented and the conclusions of this paper are well supported by data. I am in support of publication and only have relatively minor comments:

We thank the reviewer for their positive feedback on our work. We decided to combine our work into one publication to bridge experimental cell biology and experimental research on active colloids. We believe that the connection build in this manner is valuable for both communities.

Comments:

Figure 2/3: When the authors placed "multiple balls" in circular confinements, they observe the system transitions to ordered collective rotation. With three balls, the system toggles between order and disorder. Above 82 balls, order can be lost. What is the minimum number of balls where the system displays consistently ordered rotations? Are there similar constraints for cell number?

This depends on the domain size, in the correspondingly small confinement the minimum number of balls accounts to two. The reason why the three-ball system alternates between order and disorder is because at low ball numbers, the collective is more sensitive to variations within the system and therefore more susceptible to

perturbations. At increased numbers, this sensitivity is dampened by the inertia of the collective. However, with higher densities the compensating effect of inertia vanishes due to the reduced spacing between the individual balls. At low numbers or high densities an initial preselection of similar agents will keep them in the ordered state indefinitely.

Based on the observation that cells develop the rotational motion easiest when grown from a low densities, it appeared to us that similarity between cells is also essential. We believe that this results from the increased similarity between cells that stem from the same mother cell. This similarity is most prominent right after a division where the new pair of cells are roughly symmetric.

Initially starting this project, we made this observation in small circular confinements with 50 μm diameter. Immediately after division, a pair of cells rotates roughly 4 times faster than a pair that was seeded together. Eventually, both cell pairs converge to roughly similar mean speed. **(Fig. 4, Response Movie)**

Published results seem to have come to a similar conclusion. The transition to the carousel-like motion appears to be impaired when cells are seeded at high initial densities. (Doxzen K., et. al 2013) This impairment may result from the heterogeneity of the population which can be reduced when cells are grown from a low density within a single confinement.

Fig. 4 | Rotational movement of cell pairs

The mean in rotational speed over time of two cell pairs in circular confinements. ($D=50 \mu\text{m}$) The newly divided cells (split) initiate the rotation at a roughly 4 times larger mean rotation speed before converging to the mean rotation speed of the seeded cell pair (seed).

Figure 4c, Movie 5: This last "Act" of Movie 5 is likely meant to be compared to the last "Act" of Movie 3 but it would be best to have a control for Movie 5 with no treated cells but cells that have also been dyed.. The authors should also describe in more detail how

they determined that the treated cells were “passive individuals that were dragged around by their active neighbors.” Tracking/quantification would be appropriate here.

A population labelled with TAMRA shows no indication of a decreased capability of establishing the collective rotation. **(Fig. 5a, Response Movie)**

Unconfined, endothelial cell migration mimics a random walk, the MycB treatment strongly reduces the net displacement and strongly decreases migration speed. In circular confinements treated cells fail to coordinate and neither migrate effectively nor establish a rotating motion. **(Fig. 5b, Response Movie)**

Fig. 5 | Control experiments regarding the applied drug treatments underline the specificity for our purpose.

a, The TAMRA-stained cell populations establish collective rotation in circular confinements ($D = 150 \text{ um}$, $n = 10$). **b**, MycB treated cells cease to migrate efficiently, their maximal speed is strongly reduced and fail to establish collectivity in circular confinements ($D=159 \text{ um}$, $n=8$). Above we show sample trajectories for each condition respectively and over a time window of 4 h. While unconfined endothelial cells migrate randomly, when treated with MycB their migration behavior mimics a Brownian particle and the explored space is strongly reduced. In confined conditions a population of treated cells exhibit no collective migration and individuals reside at one location.

Figure 1 and Figure 4c: It is not clear why collective cell migration assays are performed sometimes in rings and sometimes discs. The authors should explain the rationale for each of these geometries.

We wanted not only to see whether but how synchronization emerges. Here, the rationale for using rings instead of circles was to have a clear contact area between precisely two cells. Whereas in circles cells tend to develop multiple and broader contact sites between multiple cells. This renders the analysis difficult and the process visually overwhelming and unclear.

Also the authors should consider repeating the results in Fig. 4c in rings where the 1D nature would allow for a more controlled forum to test the effect of a disruptor cell on collective motion.

Inhibiting the motion of a subset of cells in a ring will lead to jamming. While in circular confinements uncooperative cells are mostly bypassed, this is not possible in a ring. The 1D nature of a ring necessitates that the untreated population has to overcome the adhesive forces of the uncooperative cells and push or drag them along with the moving collective. Endothelial cells form strong adhesive forces with the substrate. The magnitude of these forces out scales the magnitude of the generated pushing forces.

Reviewer #3 (Remarks to the Author):

In their work, Riedl et al. investigate a novel mechanism that could be responsible for the coordination of collective migration of cells. To this end, they utilize fluorescence microscopy to observe a system of human endothelial cells in circular or ring-shaped confinements. Focusing on the actin cortex which has been labeled with LifeAct-GFP, the authors observe dynamic waves of actin polymerization. These waves fall in line with observations in previous studies, as they behave like excitation waves, can create spiral patterns and show velocities of a similar magnitude.

The novelty in the present study lies in the fact that the work finds a clear correlation between the synchronization of wave nucleation events and the collective turning motion of cells within a circular or ring-shaped confinement. This suggests an intriguing mechanism which enables cells to coordinate with their nearest neighbors solely due to oscillations in their actin cytoskeleton at shared interfaces. A mechanical – rather than biochemical – coupling process like this has not been described in detail before, and as such provides a valuable and very important contribution to the field.

The experimental data presented by the authors is of high quality and allows for a quantitative analysis of actin wave dynamics and cell turning at the same time, which is not trivial considering the different length- and time-scales of the two processes.

Furthermore, the authors introduce a simplified model consisting of hollow balls that are driven by an internal motor that is rotating an unbalanced weight. This system is used as an analogy, showing the results of an established physical synchronization process. Both studied systems share the same characteristic behaviours, hinting again at a synchronization of actin polymerization waves as a source of collective migration.

Because of these points, I want to recommend this work for publication, as I think that it will be a good fit for this journal. However, before it is submitted to a wider audience, I would like a few clarifications as well.

Major questions and remarks:

Most of them are related to the direct analogy between the motor-driven ball and the cell, which is not an immediately obvious one to make. The ball is directly driven by the unbalanced weight on the motor and rigid. Once there is some sort of confinement that limits its degrees of freedom, there is an immediate feedback from the available range of motion to the possible orientation and spinning frequency of the motor.

In a cell, this is not as straightforward. The waves are generated and deform the outline, but it deforms wildly even in the absence of waves. On top of that, cells can propagate via other means, as the CK666 experiments show. Other studies in different cell types go in a similar direction and identify polymerization waves mostly as a polarizing mechanism for the cell, which would also explain the results in this paper. Once the waves have polarized the cells in the cluster, those would follow the laws of collective motion of classic active particles (assuming e.g. constant and equal speeds), which have also been

found to engage in collective turning motion in circular confinements only in the presence of next-neighbor interactions.

All of this does not affect the validity of the individual results for the synchronization, it is more related to the physical origin of the effect. Currently, there is no clear distinction between coupled oscillators and polarized active particles, or even something more abstract. Depending on this, the model system would be more of a qualitative comparison to illustrate the important characteristics, rather than a simplified equivalent. While doing all of the necessary tests might be out of scope for this particular work, it is still something that should be noted.

We appreciate the reviewer's valuable remarks and insightful comments. The analogy between the ball system and the cells highlights the important characteristics and does not serve as a simplified equivalent. The key common feature is that the populations' propulsion speed is not per se constant, but uniformizes due to synchronization reliant on nearest neighbour interactions.

Also, both our systems challenge the often taken a priori assumption of constant and uniform speeds of individual agents in the context of collective migration. An assumption commonly applied in models, which does rarely hold true for most propulsion mechanisms at fast timescales. Our systems suggest that achieving a uniform propulsion speed necessitates a uniformizing mechanism such as synchronization.

We also show that in migrating cells the nucleation frequency of waves correlates positively with the migration speed of single cells and therefore presents a more intricate function extending beyond polarization.

1.) In this context, it would be interesting to see not only the standard deviation (STD) of the wave nucleation frequency over time, but also the magnitude of the average wave nucleation frequency in the cluster itself. Does it correlate with the turning rate?

We do not find that the collective turning rate correlate in the cell cluster. However, during onset of collective motion we find that the maximum in collective rotation rate coincides with an increased nucleation frequency. However, this correlation does not persist during the remaining collective rotation.

In our study, we did not observe a correlation between the collective turning rate and the magnitude in nucleation frequency in the cell cluster that persists over the course of the collective rotation. For the shown cluster, however, in the 3 hour period during the onset of collective rotation the maximum in nucleation frequency coincides with the maximum in collective rotation rate. This correlation could be explained with our observations in single cells, however, it does not persist throughout the remaining time series unlike the correlation we find in standard deviation.

Fig. 6 | Correlation of mean frequency in a collectively rotating cluster.

The mean nucleation frequency shows a peak during the onset of the carousel-like motion, but shows no correlation during the collective rotation.

2.) In a similar sense, for a direct feedback, one would expect both the turning rate and $1 - \sigma_f$ to peak at the same time, as in Fig 2e. How significant is it that in Fig 1e, the frequencies start aligning perfectly about 3 hours after the collective motion has started?

The reviewers' notion is correct. However, contrary to velocities which can be calculated from one time frame to the next, extracting frequencies accurately requires sufficiently long time windows. The shorter the chosen time window the more inaccurate the extracted average frequency. At an average frequency of four waves per hour, we chose time windows of three hours. Therefore, the significance of the alignment in frequencies shifted by one time-window is negligible and can be dedicated to the nature of how one extracts frequencies in the first place.

3.) Is it possible to affect other parts of the system with drugs? Could e.g. inhibition of myosin with blebbistatin lead to waves in the absence of collective motion?

Inhibition of myosin with Blebbistatin is a double-edged sword in addressing synchronization in collectively moving cells. While it perturbs the coupling between cells and therefore perturbs collective motion, it also renders adhesive cells immobile and impacts the individual cells' migration.

Any readout that relies on speed of a collective has to be interpreted with care. For example in ring-like confinements, inhibition of myosin eventually leads to a complete halt of collective motion, which cannot be decoupled from the loss migration in individual cells (Jain, et al. 2020). At the same concentration of Blebbistatin (80 μM) in

circular confinements, we find that cell locomotion arrests before an effect in coordination is visible. **(Fig. 2, Response Movie)**

At lower concentrations (40uM) cell migration eventually also arrest. However, in the transient period it appears that configurations that require less coordination, e.g. a symmetric arrangement, can sustain the collective rotation for longer time periods or until perturbed by an uncoordinated individual. This is in agreement with the interpretation drawn from experiments conducted in ring-like confinements, where the rotations are more robust and the order parameter drops gradually before cell motility ceases. (Jain et, al. 2020) The susceptibility of Blebbistatin treated patterns to perturbations resulting from uncoordinated cells during migration aligns well with the involvement of myosin in the formation and strengthening of cadherin dependent junctions. (Hoelzle and Svitkina, 2011)

4.) Why is specifically the nucleation frequency the quantity that is synchronized, rather than some phase or relative position?

This is a very good question and at this point we can only speculate. First, the robust definition of a relative position or phase in a cell is a challenge in itself. Whether this is possible in general is not clear due to the morphological heterogeneity of a cell during migration. Second, it appears that the difference between the timescales of the wave and migration dynamics is sufficiently large such that the integrated dynamics of the waves rather than its instantaneous behaviour are of consequence for migration. Meaning that, while on short timescales the position or phase of a single wave may cause e.g., local deformations, it seems not possible to predict the resulting cellular movement on the longer timescales. However, the integrated behaviour of multiple, consecutively nucleated waves basically quantified through their frequency reflects in the migration speed.

Is it connected to the observation in the movie that in the turning state, waves mostly travel "back to front", while they seem to be more chaotic before the onset of motion? How representative is that notion?

We were sharing the same notion as the referee. Even to us it seemed intuitive to assume that the polymerizing actin waves which move from the rear to the front of the cell should promote migration in the same direction. This intuition would align well with recent models. (Stankevics, et al. 2019) However, the travelling direction of the wave does not correlate with the migration direction of the cell. We find multiple instances where the direction of migration and wave propagation are opposite. Therefore, it appears that the mechanisms that govern cell migration are more intricate. We speculate that waves "activate" the basal membrane during migration, an increased nucleation frequency leads to a more active membrane and hence a faster migration.

We indeed observe that the chemical oscillations underlying the actin waves possess multiple states. **(Supp. Movie 1)** How the emergent states feedback on cell locomotion is currently unclear, but might serve as a fruitful soil for future investigations.

Minor remarks:

1.) I would rephrase the introduction of the order parameter C in the main script. "The maximum / the minimum" sounds like it refers to one specific instance during the experiment where the value is at its highest/lowest. A general "maxima/minima" is clearer.

We included this suggestion into the paper.

2.) The arrows in figures 1a and 2c could be a bit bigger.

We increased the size of the arrows in the figures.

REVIEWERS' COMMENTS

Reviewer #1 (Remarks to the Author):

The authors have performed all the requested experiments and answered my concerns in a very detailed and nuanced way (which I liked), I support publication without further revision.

Reviewer #2 (Remarks to the Author):

I am satisfied with the authors' response and appreciate the authors' thorough and careful work to address my comments. I recommend publication in the manuscript's current form. I believe this work will be of great interest to those in multiple fields.

Reviewer #3 (Remarks to the Author):

With their revision, the authors have clarified all of the important points that have been mentioned, while also providing new insights with new experiments. The analogy between cells and toy system is clear in this version, and as a result, I am now very much in favor of publication. I support the authors' claim that this work will be of great interest for two different scientific communities.

Revision 2

REVIEWERS' COMMENTS

We want to express our gratitude to all reviewers for their open mindedness, their appreciation for our unconventional approach and their constructive comments – rendering this review process a rare pleasure.

Reviewer #1 (Remarks to the Author):

The authors have performed all the requested experiments and answered my concerns in a very detailed and nuanced way (which I liked), I support publication without further revision.

We thank the review specifically for their constructive questions and suggestions, which have revealed potential for further investigations.

Reviewer #2 (Remarks to the Author):

I am satisfied with the authors' response and appreciate the authors' thorough and careful work to address my comments. I recommend publication in the manuscript's current form. I believe this work will be of great interest to those in multiple fields.

We thank the reviewer for their comments and specifically their words of appreciation, they have been a source of motivation.

Reviewer #3 (Remarks to the Author):

With their revision, the authors have clarified all of the important points that have been mentioned, while also providing new insights with new experiments. The analogy between cells and toy system is clear in this version, and as a result, I am now very much in favor of publication. I support the authors' claim that this work will be of great interest for two different scientific communities.

We thank the reviewer for their very insightful comments, which align with our interests and should the opportunity arise, we would like to discuss them further in detail.